# Factors Related to Hemifacial Spasm Recurrence in Patients Undergoing Microvascular Decompression—A Systematic Review and Meta-Analysis

**DOI:** 10.3390/brainsci12050583

**Published:** 2022-04-29

**Authors:** Grazia Menna, Marco Battistelli, Alessandro Rapisarda, Alessandro Izzo, Manuela D’Ercole, Alessandro Olivi, Nicola Montano

**Affiliations:** 1Department of Neuroscience, Neurosurgery Section, Università Cattolica del Sacro Cuore, 00168 Rome, Italy; mennagrazia@gmail.com (G.M.); marco.battistelli23494@gamil.com (M.B.); alessandro.olivi@policlinicogemelli.it (A.O.); 2Department of Neurosurgery, Fondazione Policlinico Universitario Agostino Gemelli IRCCS, 00168 Rome, Italy; alerapi91@gmail.com (A.R.); izzo.alessandro88@gmail.com (A.I.); manuela.dercole@policlinicogemelli.it (M.D.)

**Keywords:** hemifacial spasm recurrence, microvascular decompression, lateral spread response

## Abstract

There is a lack of knowledge about the factors associated with the recurrence of hemifacial spam (HFS) following an initially successful microvascular decompression (MVD) surgery. The aim of the present study was to systematically review the pertinent literature and carry out a meta-analysis of factors linked to HFS recurrence in patients undergoing initially successful MVD treatment. An online literature search was launched on the PubMed/Medline and Scopus databases. The following data were collected: sex, age at surgery, affected side, reported improvement after surgery, presence of post-operatory facial weakness, symptom duration, offender vessels, and data obtained from intraoperative neurophysiological monitoring. Upon full-text review, eight articles were included, studying 1105 patients, of which 64 (5.7%) reported recurrence after MVD surgery for hemifacial spasm. There was a statistically significant increased incidence of HFS recurrence in patients with the persistence of lateral spread response (LSR after surgery (OR 9.44 (95% CI 1.69–52.58) *p* 0.01), while those patients experiencing a shorter disease duration before going to surgery were significantly less prone to experiencing disease recurrence (OR 0.11 (95% CI 0.03–0.46) *p* 0.002). The remaining examined factors did not result as significantly associated with the risk of recurrence. The funnel plots were largely symmetrical for each variable studied. Taken together, the results of our meta-analysis seem to suggest that short-term symptom duration is a protective factor against HFS recurrence after MVD surgery, while LSR persistence is a negative prognostic factor. Well-designed randomized controlled clinical trials with a long follow-up are expected to further explore therapeutic alternatives for HFS recurrence.

## 1. Introduction

Hemifacial spasm (HFS) is a movement disorder characterized by either brief or persistent, intermittent twitching of the muscle innervated by the facial nerve. Contractions start from the orbicularis oculi; over time, they spread and begin to involve the lower half of the face, unilateral involuntary tonic or clonic contractions, which persist even during sleep. Etiologically, HFS can be classified as either primary, when related to a neurovascular conflict with an offender vessel (anterior inferior cerebellar artery (AICA), superior cerebellar artery (SCA), vertebral artery (VA), or posterior inferior cerebellar artery (PICA)) or secondary, due to demyelination, Bell’s palsy, cerebellar-pontine angle (CPA) tumors, arteriovenous malformations (AVM), infections, parotid tumors, or structural abnormalities. The true prevalence of the disease is unknown due to the large number of under- and misdiagnoses: of the two existing forms, epidemiological studies usually focus on the primary type. The mean prevalence of the latter is 11 around per 100,000 total population, while familial and bilateral cases are rare [1,2,3]. This condition has a major impact on affected patients: it is linked to poor quality of life, due to disturbed sleep and insomnia, social embarrassment, and functional blindness. Being a chronic condition, with progressively increasing spasms, therapeutic options comprise drugs (such as anticonvulsants), botulinum toxin, or microvascular decompression (MVD) surgery. Pharmacological treatment (clonazepam, carbamazepine, gabapentin) is usually not preferred, due to the heavy side effects and for not being effective in solving the condition. Botulinum toxin treatment, mainly with onabotulinumtoxinA, often represents a temporary solution that needs repeated injections. Surgery is indicated in case of primary disease due to neurovascular conflict (NVC). The MVD success rate is generally higher than 90%, even in long-term follow-up reports. However, there have been reports of late or delayed recurrence of facial spasm following an initially successful MVD, a phenomenon already described for trigeminal neuralgia (TN), which has pathogenesis and surgical response similar to HFS. Currently, evidence is lacking about the factors linked to delayed recurrence of HFS after initially successful MVD [4,5,6]. Although the difficulties in treatment are shared by HFS and TN, comprehensive European guidelines have recently been released only for the latter [7], and there is a lack of evidence for HFS. Thus, improving the understanding of factors associated with the prognosis may be relevant to improving the management of HFS patients. Therefore, the aim of the present study was to systematically review the pertinent literature and carry out a meta-analysis to identify clinical, radiological, and intraoperative neurophysiological data linked to HFS recurrence in patients undergoing initially successful MVD treatment.

## 2. Materials and Methods

### 2.1. Review Question

The herein presented study was conducted in accordance with the PRISMA-P (preferred reporting items for systematic review and meta-analysis protocols) guidelines [8].

The review question was formulated according to the PICO (P: patients; I: intervention; C: comparison; O: outcomes) guidelines: In patients suffering from hemifacial spasm (P) who underwent MVD surgery (I) is it possible to identify factors (C) linked to an increased risk of recurrences (O)?

### 2.2. Inclusion Criteria and Outcome Measure

An online literature search was launched on PubMed/Medline and Scopus databases using the following research string: “((Facial OR Hemifacial)) AND (Spasm) AND (Microvascular decompression OR MVD) AND (Outcome OR Recurrence))”; the latest research was conducted in December 2021. Two authors (M.B. and G.M.) independently conducted the abstract screening for eligibility. Any discordance was solved by consensus with a third, senior author (N.M.). No restrictions on date of publication were made. Only comparative studies detailing a clear description of patients reporting recurrence after MVD for HFS were included. HFS recurrence was defined as recurrence of facial spasm following initially successful MVD treatment; the timing of recurrence was reported as variable; both early (less than 1 year) and late (more than 1 year) recurrences were included. HFS recurrences also included early spasm reappearances after an initially successful MVD but did not include transient contraction of the orbicularis oris occurring in some patients after MVD. Exclusion criteria were as follows: no comparative study design, studies without recurrence and no recurrence comparison, studies published in languages other than English, and meta-analyses. A systematic abstract screening of the references (forward search) was performed, in order to identify additional records.

The following data were collected for meta-analysis, in order to evaluate their role in hemifacial spasm recurrence: −Demographic and clinical data: sex, age at surgery, affected side, reported improvement after surgery, presence of post-op facial weakness, symptoms duration (less or more than 2 years);−Radiological data: offender vessel (AICA, PICA, VA, veins, multiple vessels).−Intraoperative neurophysiological monitoring data: lateral spread response (LSR) (if disappearing or being present after surgery).

### 2.3. Statistical Analysis

Statistical analyses were performed using Review Manager (RevMan) (Version 5.4, The Cochrane Collaboration, 2020, London, UK) applying the random effect model. Heterogeneity was tested using the chi-square test and quantified by calculating the I^2^ statistic, in which *p* < 0.05 and I^2^ > 50% were considered statistically significant. For the pooled effects, odds ratio (OR) was calculated for dichotomous variables and weighted mean difference (WMD) was calculated for continuous variables. Continuous variables are presented as mean differences and 95% confidence intervals (CI), whereas dichotomous variables are presented as ORs and 95% CI. Publication bias was tested using a funnel plot.

## 3. Results

### 3.1. Systematic Review

The search of the literature yielded a total of 1480 results. Duplicate records were then removed (*n* = 30). A total of 1450 were screened, and 1403 records were excluded via title and abstract screening; 47 studies were found to be relevant to our research question and were assessed for eligibility (Figure 1). Upon full-text review, 8 articles were included in the review, including 1105 patients (Table 1), of which 64 (5.7%) reported recurrence after MVD surgery for hemifacial spasm [9,10,11,12,13,14,15,16,17].

### 3.2. Meta-Analysis

The meta-analysis results are summarized by Table 2, Figure 2, Figure 3 and Figure 4. Among the available factors, symptom duration more than 2 years (OR 0.11 (95% CI 0.03–0.46) *p* 0.002) (I^2^ = 0%, *p* 0.39) and LSR persistence after surgery (OR 9.44 (95% CI 1.69–52.58) *p* 0.01) (I^2^ = 28%, *p* 0.25), each analyzed in three out of the eight papers included, resulted as significantly associated with HFS recurrence after MVD treatment. The remaining examined factors did not result as significantly associated with HFS recurrence: demographic, clinical, and radiological data (sex, age at surgery, affected side, reported improvement after surgery, presence of post-op facial weakness, offender vessel) had no significant impact on the outcome, despite being reported in most of the studies. Additionally, LSR disappearance during surgery not was found to be statistically significant (see Table 2). Regarding the observed effects, the funnel plots suggest that publication bias did not play a significant role (Figure 2, Figure 3 and Figure 4) in the retrieved texts. 

## 4. Discussion

### 4.1. Hemifacial Spasm: An Overview 

HFS is characterized by involuntary clonic and/or tonic contractions of the muscles of facial expression, usually unilaterally, beginning in the periorbital musculature and later progressing [1,2,3]. The condition is usually due to the presence of an ectatic or aberrant blood vessel, which compresses the facial nerve at this root entry/exit zone, leading to local demyelination. There are two theories that explain how compression leads to HFS: the peripheral theory, and the nuclear theory. The first states that an ephaptic transmission of impulses between neighboring neurons could lead to an abnormal firing; the second is focused on facial nucleus hyperexcitability due to an irritative feedback from a nerve peripheral lesion. The spasms can be brief and localized or—more often—can occur in bursts, contracting the whole hemiface in a more tonic manner, resulting in a disfiguring grimace in severe cases, and causing severe discomfort to the patient. Primary HFS diagnosis requires magnetic resonance imaging (MRI) to exclude a secondary hemifacial spasm and, second, to search for and characterize the NVC. The most frequent vascular compressions are from the AICA, PICA, and VA; venous conflicts are very rare, while multiple ones are not so infrequent. The condition is also usually aided by some anatomical variability: arterial dolichoectasia, posterior fossa with a small volume, or bony malformations. From a neurophysiological point of view, the HFS hallmark is represented by the LSR: the blink reflex spreads to muscles other than the orbicularis oculi, probably due to antidromic impulse transmission between neighboring fibers of the facial nerve, due to ephaptic transmission. Many diseases enter into the differential diagnosis of HFS, such as blepharospasm, tardive dyskinesia, psychogenic hemifacial spasm, aberrant regeneration after facial nerve injury, motor tics, and focal cortical seizures involving facial muscles. Due to the low prevalence of the disease, there have been no clinical randomized trials to determine the best available treatments. Botulinum toxin treatment offers a simple, non-invasive therapy for this condition, even if temporary. The main drawbacks of the botulinum toxin are related to the cost of therapy and the repeated injections every 3–6 months. Due to its great efficacy and minimal side effects, botulinum toxin is considered the first therapeutic modality for HFS management. MVD is accepted as the gold standard in the management of resistant primary HFS, due to the high success rate of the procedure, the rarity of serious complications, and low mortality rates. However, the procedure involves risks, due to the interference from a very small area, the presence of critical anatomical structures, and a recurrence rate ranging between 2.1–7.3% [18,19,20]. Failing surgical measures, there is no consensus on recurrent HFS treatment. Therefore, our study aimed to identify factors linked to HFS recurrence in patients undergoing initially successful MVD treatment. We believe that a correct identification of the causes of recurrence could help in the management of these patients [13,14,15].

#### 4.1.1. Demographic and Clinical Data

Regarding clinical data, they were variously reported in the examined studies. Our results show that demographic data (age and sex) did not play any role in recurrence, even if HFS is more frequently observed in females than males. Focusing on the proper clinical data, the absence of proper assessment of additional clinical risk factors should be noted: for example, the presence of hypertension in affected patients was examined in only one study of the eight included. Narrowing down the attention to those factors strictly linked to facial nerve function, it is interesting to notice how an immediate improvement pattern or, on the contrary, postoperative weakness of the affected nerve, were scarcely and selectively reported, and not resulting as related to the recurrence [10]. In contrast, pre-operatory symptom duration was the only clinical factor statistically related to outcomes: our study showed that in patients with less than two years of symptom duration, HFS recurrence after MVD was significantly reduced. This is of extreme relevance from a practical point of view. First, it suggests the need to refer to surgery patients with poor responsiveness to conservative treatments and with short disease duration; second, it suggests that patients should be informed of the possibility of MVD at an early stage, since delayed surgical treatment could be related to an increase risk of recurrence [16].

#### 4.1.2. Radiological Data

The various offending vessels in HFS showed no significant difference in terms of the relationship with outcome after MVD surgery. This parameter was ubiquitously reported in five out of eight of the examined papers, and, therefore, was included in the meta-analysis (Table 2). However, other radiological parameters, even if separately reported only in two of the examined studies, seemed of interest and deserve further discussion. In their study, Zhu et al. performed a comparison between the neuroimaging parameters on the affected and unaffected sides in HFS patients [7]. They considered parameters such as facial nerve angle, defined as the angle between medial margin of the acoustic-facial bundle and the anterior surface of the pons at the root entry zone (REZ); cross-sectional area of CPA cistern, defined as the region between the posterior surface of the petrous bone and the anterior surface of the pons and cerebellum, filled with cerebrospinal; and length of the cistern segment of the facial nerve, which was measured in the axial image from the point of pons where the nerve emerges from to the cochlea foramen in the internal acoustic meatus. They demonstrated how the small facial-nerve angle and small cross-sectional CPA cistern area might increase the incidence of HFS and how those factors might be used to predict the long-term outcome of MVD. Zhao et al. focused instead on the morphological characteristics of posterior cranial fossa (PCF) [12]. Currently, the limited reports have indicated that PCF crowdedness negatively influences the outcome of MVD surgery for HFS. In their study, although PCF volume seemed to be larger in HFS patients with good outcomes, this difference was not statistically significant. However, they showed how a flat-shaped PCF may be related to poor long-term outcome after MVD for primary HFS, increasing the opportunity for HFS recurrence. Although there was no room to inscribe these pioneering data in the meta-analysis, the aforementioned factors will likely be objects of increasing interest, with their actual role in HFS recurrence being clarified soon.

#### 4.1.3. Intraoperative Neurophysiological Data

The pathogenesis of HFS may involve vascular compression of the facial nerve at its origin near the brainstem, resulting in demyelination and ephaptic transmission. LSR is defined as a pathological latent, abnormal response elicited by the stimulation of one branch of the facial nerves of patients with HFS, resulting in the contraction of the facial muscles innervated by the other branch of the facial nerve, probably due to cross-transmission of antidromic activity from the stimulated branch of the facial nerve contributing to the phenomenon [21,22]. During MVD surgery, continuous monitoring of the LSR is used to confirm that adequate decompression has been performed. Although previous studies [16] demonstrated a positive correlation between the decompression of the facial nerve and the intraoperative disappearance of LSR, our meta-analysis failed to demonstrate this association. On the other hand, persistence of LSR at the end of surgery appears to be more significant in predicting HFS recurrence after MVD. These data suggest that LSR disappearance might be more useful in identifying the vessel responsible for NVC, rather than in predicting the outcome. The persistence of LSR at the end of surgery might be related to an inadequate decompression of facial nerve or the presence of another vessel involved in the NVC. In any case, a further exploration of facial nerve is advised if there is a persistence of LSR, because this exposes the patient to a greater risk of recurrence.

#### 4.1.4. Limitations

Our study has some limitations: the heterogeneity among the included studies is variable outcome by outcome, occasionally ranging very high; some of the data were not available for meta-analysis; there is a lack of a standardized method of monitoring and interpreting LSR in the considered studies. In addition, the low number of included studies suggests that the current knowledge about HFS recurrence is relatively lacking.

## 5. Conclusions

Taken altogether, the results of our meta-analysis suggest a statistically significant increased incidence of HFS recurrence in patients with persistence of LSR after surgery, while those patients experiencing a shorter disease duration before undergoing MVD were significantly less prone to experiencing disease recurrence [23,24,25,26]. However, the low number of included studies suggests that the current knowledge about HFS recurrence is scarce and insufficient to aid in clinical practice [26,27,28,29,30,31,32]. More studies are needed to draw significant conclusions that may help in the identification of patients at higher risk of HFS recurrence, and who may benefit from different approaches. Given the high social yield of the disease and the absence of a consensus in HFS recurrence treatment, well-designed randomized controlled clinical trials with a longer follow-up are needed to further explore therapeutic alternatives for HFS recurrence.

## Figures and Tables

**Figure 1 brainsci-12-00583-f001:**
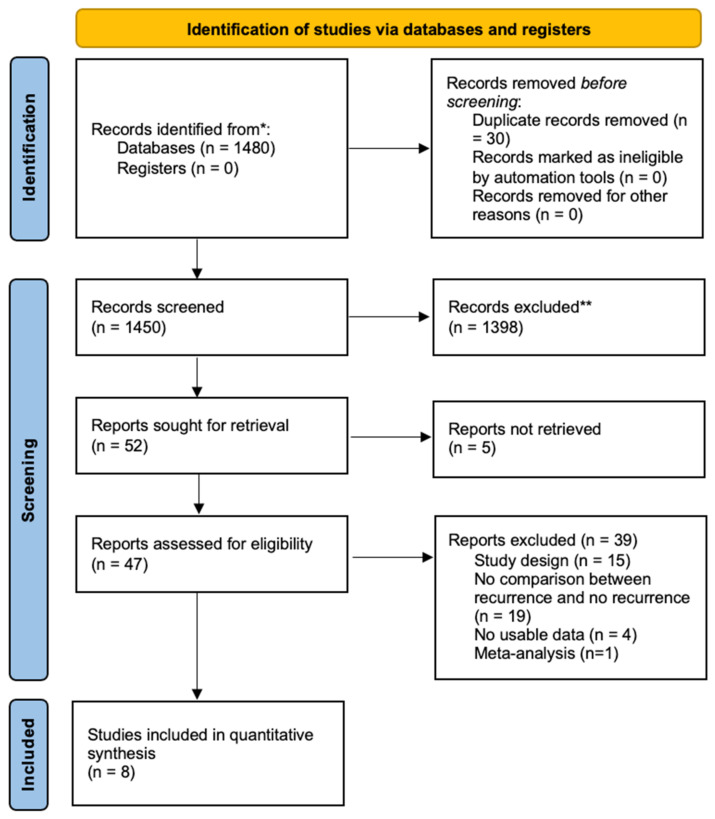
Systematic review flow diagram. The PRISMA flow diagram for the systematic review, detailing the database searches, the number of abstracts screened, and the full text.

**Figure 2 brainsci-12-00583-f002:**
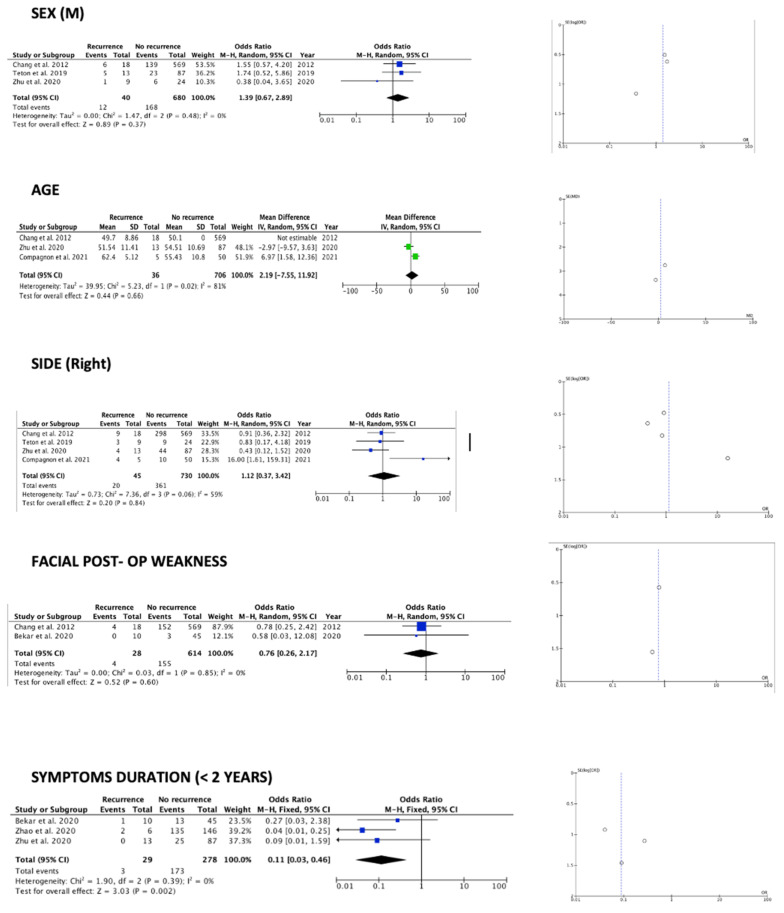
Forest plots and funnel plots for investigated clinical variables [10,11,13,14,15,16].

**Figure 3 brainsci-12-00583-f003:**
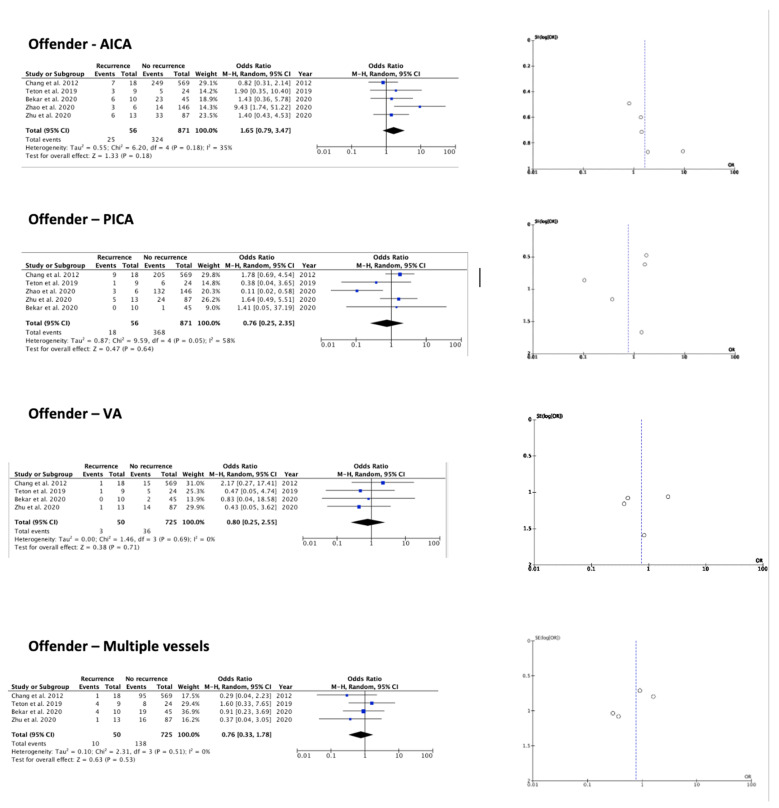
Forest plots and funnel plots for investigated radiological variables [10,11,13,14,15,16].

**Figure 4 brainsci-12-00583-f004:**
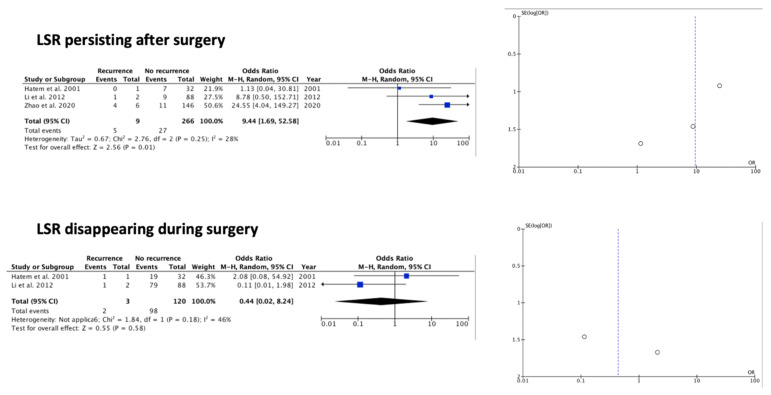
Forest plots and funnel plots for investigated intraoperative neurophysiological data. LSR (lateral spread response) [9,12,16].

**Table 1 brainsci-12-00583-t001:** Summary of studies included in the meta-analysis.

Name	Type of Study	Total Number of Patients (Recurrence/No Recurrence)
Hatem et al., 2001 [9]	Prospective	33 (1/32)
Chang et al., 2012 [10]	Retrospective	587 (18/569)
Li et al., 2012 [12]	Retrospective	90 (2/88)
Teton et al., 2019 [13]	Retrospective	33 (9/24)
Bekar et al., 2020 [14]	Retrospective	55 (10/45)
Zhao et al., 2020 [16]	Retrospective	152 (6/143)
Zhu et al., 2020 [11]	Prospective	100 (13/87)
Compagnon et al., 2021 [15]	Retrospective	55 (5/50)

**Table 2 brainsci-12-00583-t002:** Meta-analysis of possible factors involved in hemifacial spasm recurrence.

Factor	Or	95% CI	*p* Value	I^2^	I^2^ *p* Value
Demographic and clinical data					
Sex (M)	1.39	0.67–2.89	0.37	0	0.48
Mean age at surgery (years)	2.19	−7.55–11.92	0.66	81%	0.02
Side (Right)	1.12	0.37–3.42	0.84	59%	0.06
Facial post-op weakness	0.76	0.26–2.17	0.6	0	0.85
Symptoms duration (<2 years)	0.11	0.03–0.46	0.02	0	0.39
**Radiological data (offender vessel)**					
AICA	1.65	0.79–3.47	0.18	35%	0.18
PICA	0.76	0.25–2.35	0.64	58%	0.05
VA	0.80	0.25–2.55	0.71	0%	0.69
Multiple veins	0.76	0.33–1.78	0.53	0%	0.51
**Intraoperative neurophysiological data**					
LSR disappearing during surgery	0.44	0.02–8.24	0.58	46%	0.18
LSR present after surgery	9.44	1.69–52.58	0.01	28%	0.25

## Data Availability

Not applicable.

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
