# Peer review of "Factors Related to Hemifacial Spasm Recurrence in Patients Undergoing Microvascular Decompression—A Systematic Review and Meta-Analysis"

_brainsci, 2022, doi:10.3390/brainsci12050583_

Round 1

Reviewer 1 Report

The study results are interesting. I would ask the authors to clarify: 
How did you define HFS recurrence? The transient contraction of the orbicularis oculi occurs in some patients after MVD. Was this contraction also included in the HFS recurrence, or was a spasm that required further intervention included in the recurrence? Was a spasm that reappear early after initially successful MVD  (within 1 week after MVD) included in the recurrence?

Author Response

Dear Reviewers,

We thank you for the positive comments on the paper and for your detailed review. We are glad that this manuscript was of interest to you. Following your carefully structured observations and recommendations, the manuscript was modified. Below is a point-to-point response to your comments.

 Reviewer #1

  • How did you define HFS recurrence? The transient contraction of the orbicularis oculi occurs in some patients after MVD. Was this contraction also included in the HFS recurrence, or was a spasm that required further intervention included in the recurrence? Was a spasm that reappear early after initially successful MVD (within 1 week after MVD) included in the recurrence?

We thank the reviewer for the valuable and constructive comments. In studies included in our meta-analysis, HFS recurrence was defined as recurrence of facial spasm following initially successful MVD treatment; the timing of recurrence was reported as variable; both early - less than 1 year – and late – more than 1 year- recurrences were included. Therefore, HFS recurrences included also early spasm reappearances after initially successful MVD but did not include transient contraction of the orbicularis oris occurring in some patients after MVD, as now specified in the manuscript.

Reviewer #2

  • This is a well-written systematic revie paper. I just find a typo "usually" in Line 214.

We thank the reviewer for the positive comments. We corrected the typo as requested.

Reviewer 2 Report
This is a well-written systematic revie paper. I just find a typo “usually” in Line 214.

Author Response

(The authors gave the same response as above.)
